# Placing sensors in sewer networks: A system to pinpoint new cases of coronavirus

**Mehdi Nourinejad** [1] *, **Oded Berman** [2], **Richard C. Larson** [3]

**1** Civil Engineering Department, York University, Toronto, Canada, **2** Rotman School of Management, University of Toronto, Toronto, Canada, **3** Massachusetts Institute of Technology, Cambridge, Massachusetts, United States of America

* mehdi.nourinejad@lassonde.yorku.ca

**Data Availability Statement:** The data used in this study are third party data from Mapsonline.net and can be accessed following the protocol outlined in the Methods section.

## Abstract

We consider a proposed system that would place sensors in a number of wastewater manholes in a community in order to detect genetic remnants of SARS-Cov-2 found in the excreted stool of infected persons. These sensors would continually monitor the manhole's wastewater, and whenever virus remnants are detected, transmit an alert signal. In a recent paper, we described two new algorithms, each sequentially opening and testing successive manholes for genetic remnants, each algorithm homing in on a neighborhood where the infected person or persons are located. This paper extends that work in six important ways: (1) we introduce the concept of in-manhole sensors, as these sensors will reduce the number of manholes requiring on-site testing; (2) we present a realistic tree network depicting the topology of the sewer pipeline network; (3) for simulations, we present a method to create random tree networks exhibiting key attributes of a given community; (4) using the simulations, we empirically demonstrate that the mean and median number of manholes to be opened in a search follows a well-known logarithmic function; (5) we develop procedures for determining the number of sensors to deploy; (6) we formulate the sensor location problem as an integer nonlinear optimization and develop heuristics to solve it. Our sensor-manhole system, to be implemented, would require at least three additional steps in R&D: (a) an accurate, inexpensive and fast SARS-Cov-2 genetic-remnants test that can be done at the manhole; (b) design, test and manufacture of the sensors; (c) in-the-field testing and fine tuning of an implemented system.

## 1. Introduction and overview

The Coronavirus disease (COVID-19, caused by the virus SARS-CoV-2) is highly infectious and dangerous. The best disease control is infection avoidance: having all members of society wear masks, do social distancing, wash hands frequently and undertake other beneficial behavioral and hygienic steps. Despite controls, some people will become infected. A newly infected person may show no symptoms and not even be aware of his/her infection for up to one or even two weeks; during this initial period, the newly infected individual may become highly contagious. Such asymptomatic infectiousness is one of the confounding properties of the

**Funding:** This study was partially funded by the Natural Sciences and Engineering Research Council of Canada (NSERC) to OB. No additional external funding received for this study.

**Competing interests:** The authors have declared that no competing interests exist.

Coronavirus and the motivation for frequent monitoring within the population. Early identification of newly infected people will greatly reduce disease spread.

We build on the new field of Wastewater-Based Epidemiology (WBE),

". . .WBE postulates that through the analysis of population pooled wastewater, infectious disease and resistance spread, the emergence of new disease outbreak to the community level can be monitored comprehensively and in real-time." [1]

With COVID-19, the majority of infected individuals excrete remnants of the SARS-Cov-2 virus (viral RNA—ribonucleic acid) in their stool, into the toilet and then into the municipal sewage system [2–4].

Using WBE, detecting such virus remnants in the raw sewage is proof that one or more persons "upstream" from the detection point are infected with COVID-19. This virus-remnants-in-human-stool usually starts very early in the infection period, well before visible symptoms, typically giving a one week "early warning" of the infection. With WBE, the "invisible" becomes "visible." In the year 2020, multiple researchers in different countries verified this early warning finding by collecting and testing raw sewage entering Wastewater Treatment Plants (WTP) and comparing the results to disease statistics collected later [5–7].

In the fall of 2020, using the idea of pooled testing, universities started using WBE to discover new Coronavirus infections in student dormitories. Led by the University of Arizona, now scores of universities (including MIT, University of Kentucky and the University of New Hampshire) monitor and test the wastewater from manholes adjacent to student dormitories, these manholes transiting all toilet flushes from the dormitories [8]. When the "Red Light" signal appears of a new infection in a dormitory previously free of infections, university health professionals test each student in the dormitory, identify the one(s) infected, isolate and move the individual(s) to medical care. Contact tracing then commences.

As this paper is written early in 2021, we seek to move beyond student dormitories and to develop a method of finding in a community of many thousands a newly infected patient (called "Newbie") or an infection "Hot Spot" with many infected people.

Other researchers are also now studying sewage sludge with mathematical models to make inferences about prevalence of the disease. For example, using a mathematical epidemic model, Kaplan et al. [9] examine time series of SARS-CoV-2 RNA in municipal sewage sludge vs. times series of local hospital COVID-19 admissions. They find good correspondence, with sewage sludge measurements typically 3 to 5 days ahead of hospital admissions. They draw several other inferences from the work, including a new accurate method to estimate the basic reproductive number of the disease, $R_0$.

COVID-19 has become a disease that can be tracked with the tools of WBE. And it has also become a disease that can be attacked with data and mathematical models.

## 2. Framing the problem and reviewing the literature

Our recent paper in *PLOS ONE*, "Sampling manholes to home in on SARS-CoV-2 infections" [5], provides additional background and motivation, as well as building blocks that we use here. We have written this new paper to be self-contained, and we extend earlier results in important ways. Our new model is more detailed, more realistic than that described in our previous paper. The added realism is to benefit those who may want to implement our methods in their own communities. And, most importantly, we extend past results to designing a system having in-the–field sewage-monitoring sensors.

Any municipal sewage system is a network of sewage-transporting pipes leading from all the possible originating sources in the community to the final destination, the WTP. This network has no cycles. It is a ***tree network*** whose topology is known and that we exploit in our work. We describe this network in detail in Section 2.2.

## 2.1. Sampling manholes—Review

Our dynamic algorithmic work focuses on manual sampling and testing the sewage from manholes, each a "node" in the sewer system tree network. A key observation is this: If the sewage from a given manhole tests positive for SARS-CoV-2, then there are one or more infected individuals upstream from that manhole. If that sewage tests negative, then no individual upstream from that manhole is infected. This is assuming perfect testing, an assumption we use throughout.

We start with a community that is known to be devoid of Coronavirus. While there may have been infections in the past, "today"–via recent testing—there are no known infections. Then one day at the WTP, a light flashes red: Remnants of Coronavirus are found today in the community's sewage. Chances are that the person infected is not even aware of it, due to the typical asymptomatic one-week delay from the infection event until the onset of symptoms. Yet the individual, even before symptoms, soon may become highly infectious, endangering all around him/her. Just like the student dormitory testing, the challenge is to identify the infected individual as soon as possible and move him/her into supportive isolation, thereby avoiding infection of others. Follow up with contact tracing for all of those who were in contact with the infected individual.

But suppose the community we are speaking of has 35,000 residents! We cannot test them all. We need to narrow down greatly the number of possible persons who could be infected into a small sub-population, so that it is feasible to test each in the sub-population.

We do that by sequentially testing manholes for Coronavirus. Suppose we pick as the first manhole to test one where the average sewage flow in the main pipe is about one half of its average maximum flow, the maximum occurring at entrance to the WTP. If that selected manhole tests negative, then no one upstream from this 50%-flow point (usually about 50% of residents) is infected, but someone downstream is! If that manhole tests positive, then someone upstream is infected and assuming only one person in the entire community is infected, we continue our search upstream. This process continues, each time selecting an unopened manhole that is near the "upstream-downstream" 50%-flow of remaining suspect manholes, and each time finding the direction of the source of the infection. Due to the discarding of about half of the remaining flow each time, the search space is approximately cut in half at each iteration. This means that for a community with 128 manholes, each with equal flows, we would only need about seven manhole openings and tests, since $128 = 2^7$. Our work is more general than this simple description, as we allow for spatial heterogeneity in the sources for flows.

The algorithm terminates when we have identified the "Source manhole" and know with certainty that the virus is coming from nearby sources feeding directly into this manhole. The Source manhole is the first in the system to receive sewage from the residence of Newbie. The suspect area is now the immediate sewage "catchment zone" of the Source manhole, usually part of a city block with four to eight homes feeding the sewer system into this most-upstream positive-tested manhole. If we are successful, we have reduced our population to test from 35,000 to the number of people in those homes. The U.S. Census Bureau reports that the average U.S. suburban home has 2.7 occupants [10]. For example, if we have eight homes to test, then we have only about 22 people to test. After testing and identification of the infected person(s) (who will immediately be isolated and treated), it is likely that all other individuals in

the residence of the infected person(s) would be quarantined, and contact tracing would commence. We reduce the size of the problem in a community of tens of thousands to the size of university dormitory problem, a small university dormitory!

We have just described the basis for the "Tributary Search Algorithm" of our previous *PLOS ONE* paper, referred to there as "Algorithm 1." The earlier paper also had an "Algorithm 2," used to home in on an infection 'hot spot' in a community that has a significant number of people infected.

## 2.2. Embedded sensors, the need

Some sewer systems are too large to complete all manual manhole tests in one day. According to public works experts with whom we have consulted, assuming that fast on-scene tests are available, each test would require about one hour, start to finish. That means that eight manholes tests in one day is about the maximum. Since we seek speed in identifying the infected individual(s), we want to assure that the testing can be done in one day. In this new paper, we do that by strategically placing semi-permanent sensors in selected manholes, sensors that would in real time relay any "Red Light" reporting coronavirus infection in sewage passing through that manhole. From the point of view of our sewage testing for COVID-19, each such implanted sensor acts as a local WTP. This second paper addresses that problem: How many sensors do we need so that all manhole tests could be done in one working day? And where do we place such sensors to have maximal beneficial effect?

The other reason for a second paper is the need for system realism. If our methods are to be used in practice, our model of the sewer network needs to be much more realistic than the initial prototype described in our earlier paper.

## 2.3 Embedded sensors, the state of the art

The idea of in-place sensors to detect contaminants has a rich history, both in sewage systems and in municipal water distribution systems. Beyond water and sewage systems, the field of Operations Research has a deep research history focused on determining the optimal number and locations of "flow-intercept" facilities such as sensors on networks [11, 12].

Starting with water systems, we find that in-system sensor placement has a literature going back decades. Since water is an essential fluid for sustaining life, the focus of the sensor system is on discovering quickly any unwanted contaminant intrusions into the system. Since water flows at finite speeds, placing sensors strategically within the system reduces time until detection. The mathematical analysis is often made difficult by the need to model the nonlinear system hydraulics within the water system graph. This complication results in need to expand beyond analytical models and algorithms, and to include complex simulations. An extensive and comprehensive recent overview of methods is given in Oluwaseye et al. [13]. The authors describe the approaches used and problems encountered by hundreds of authors, over a span of 50 years. The analytical methods include virtually all the same methods used by operations researchers when they study network-based systems. These include optimization algorithms, both exact and heuristic, deterministic and stochastic simulations, graph theory and more. Because of their liquid environment, the great majority of authors also must contend with modeling system hydraulics, and as mentioned above, this hugely complicates analysis and makes difficult the solution to even moderate-sized problems. An important paper not cited by Oluwaseye et al. [13] is one by Giudicianni et al. [14]. In this paper, the authors simplify the problem by not requiring any hydraulic modeling of the water distribution system, focusing only on the topology of the distribution network, mathematically a graph. They look for neighborhoods within the graph for "centrality," surrogates for so-called optimal solutions. Their

graph-theoretic intuitive approach has attributes in common with ours and with many operations research papers dealing with locations on graphs. Despite the myriad papers written on the subject of sensors in water distribution networks, significant analysis problems remain for most researchers: excessive computation time for large water-distribution networks, as the problem is described mathematically as NP hard [15]; no agreement on the best objective function; convergence only to an approximately optimal solution; and lack of significant reported impact in the field. But the field of sensors in water distribution systems is projected to grow significantly in the years ahead, perhaps to over $2 billion by 2030, when added with use of sensors in wastewater networks [16]. Because of increased need, sensor placement in water systems remains a very active research topic, and we expect to see significant advances both in theory and application in the years ahead.

Switching to sewage system applications, the historical problem is not life-threatening pollution as found with water systems, but rather continual monitoring of system flows and content to assure uninterrupted safe operation of wastewater treatment plants. In combined systems that receive storm drainage as well as usual sewage, the volume of the extra input could exceed the capacity of the wastewater treatment plant, resulting in raw effluents being spilled directly into public waters. In-system sensors can detect differences in usual flow composition and alert authorities to possible downstream complications. Sewage systems can also experience accidental spills as well as deliberate unauthorized discharges (e.g., industrial effluent), and these may also affect negatively the operation of the wastewater treatment plant. Sensors are used to detect such intrusions. Finally, as will be discussed later in this paper, the most recent intense efforts directed towards the monitoring of sewage flows deals with detecting remnants of Coronavirus in the population served by the sewage system. Regarding use of mathematical modeling to assist planners in sensor placement, the recent paper by Banik et al. [17] is representative of the class of papers using a mathematical modeling and optimization approach. The focus is to place sensors in the system to detect unwanted and possibly dangerous liquid intrusions of various types. The authors seek to locate a number of sensors within the sewage system network to achieve a combination of objectives, the main one being "time until detection." The decision variables are N, the number of sensors, and their locations. After developing their heuristic algorithmic approaches, they illustratively apply their model and algorithms to a small town in Italy. No actual implementations are reported. And the resolution of the location of the unwanted intrusion is limited to the number and placements of the sensors. There is no additional locational step after identifying the location of the first sensor to flag a new intrusion. In a second paper, B.K. Banik et al. [18] treat various modifications to the above problem, again applying simulation-algorithmic methods to that small town in Italy. The authors admit that computation time can become enormous for all but the smallest of problems. We should also mention a third paper by Banik et al. [19]. There are many additional sensor-focused papers cited in the three publications above, with heuristic problem-solving approaches similar to those described here. Finally, we need to conclude our brief overview with the fact that placing semi-permanent sensors in sewage systems is fraught with practical difficulties [20]. As these considerations may affect the success of implementing our proposed sensor placement methodology, we discuss these issues towards the end of the paper.

## 3. The sewage pipeline network

We now describe our new modeling of the tree network that depicts the connected set of wastewater underground pipes in any community. Fig 1 depicts a portion of the map of the sewer system of the Town of Belmont, Massachusetts [21]. The entire system, for a town of 26,000 residents covering 4.7 square miles, has 77 miles of sewer pipes and about 2,030

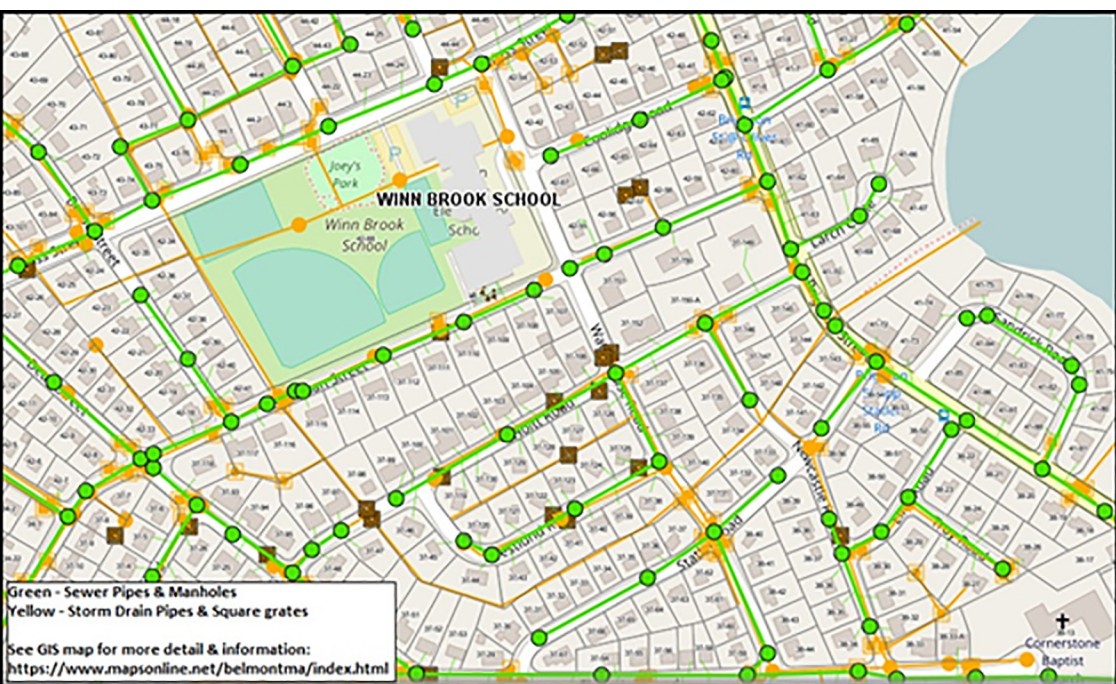

**Fig 1. Map of a portion of the underground sewer pipeline network of Belmont, Massachusetts (thanks to PeopleGIS for granting permission to publish this MapsOnline site.).**

manholes. The green line segments in the figure depict underground sewer pipes of the network. The green circles located at various points on the green sewer lines depict manholes, the eventual source of data for our algorithms. The adjacent town Lexington, Massachusetts (population 34,000, 16.5 square miles) has 4,924 manholes distributed of a 171-mile sewer pipeline network [22]. A typical American suburban town may have 50 to several hundred miles of sewer pipes, all leading to the local WTP, and between 200 and 5,000 manholes. The number of sewer manholes nationwide is about 12 million, averaging approximately 300 feet apart [23].

The sewage tree network has at least three types of pipes, each with an ascending diameter for increasing flow conditions. The three pipe types are branch sewer, main sewer and trunk sewer. The branch sewer, serving a limited number of buildings in a small geographic area collects sewage from buildings' sewers and conveys it to a main or trunk sewer. A main sewer collects sewage from two or more branch sewers acting as tributaries. The trunk sewer conveys sewage from many main and branch tributary sewers over large areas to the WTP.

## 3.1. Modeling the sewer pipeline network

The map shown in Fig 1 is typical of the sewer system maps we would see for most communities in the U.S. We use the figure to guide us on modeling the tree network of the sewer system. Our modeling involves two highly similar yet different networks: (1) the network of a community's streets and (2) the sewage pipeline tree network, all pipes assumed to be under streets. For the street network, nodes are the street intersections and links are the road segments connecting adjacent street intersections. For the sewage pipeline network, nodes are manholes and links are the sewage pipes connecting adjacent manholes. Since the under-streets sewage pipeline network is a tree network and the road network is not, it should be clear that the total

length of links in the sewage pipeline network is less than the total length of roads in the street network.

Our modeled sewage system tree network has the following additional four attributes:

1. Every street intersection contains a manhole, a sewer network node.

2. There are two types of street intersections: The three-way or T-intersection and the four-way or Crossroads intersection. With the exception of sewage pipe "dead ends," we assume that the underground sewer pipes follow the street geometries above ground. Underground, the T-intersection has two sewer pipes flowing into a third pipe; the Crossroads intersection has three sewer pipes flowing into a fourth.

3. A dead-end sewer line, possibly but not necessarily under a dead-end street segment, will have a manhole at its termination point.

4. The majority of manholes are not at street intersections or street dead ends, but on the interiors of street segments. For simplicity, we will space them evenly along the interiors of streets, with mean distance between adjacent manholes equal to the community's empirical mean (typically 150 to 300 feet).

A study of the map in Fig 1 verifies that our descriptions above summarize the key network properties of the sewer pipeline system. As always in modeling, there will be some exceptions, such as an intersection with more than one manhole. We seek not 100% accuracy but sufficient detail to capture the key attributes of the system.

### 3.2. Catchment zones

The catchment zone of a manhole "$X$" is the small collection homes, apartments, businesses and other sewage-producing facilities whose sewage encounters manhole $X$ first. Each manhole has it 'own' catchment zone. Over the community being modeled, catchment zones are mutually exclusive and collectively exhaustive.

Since the majority of manholes are in the interior of city or town blocks, most catchment zones comprise a small number of contiguous structures in the interior of the block. But catchment zones can be more complex, as shown in Fig 2, with the (red) Source manhole located at a Crossroads intersection. The infected "Newbie" individual could be located in any of the three upstream directions from the source manhole: North, West or South. In this instance, the catchment zone of the Source manhole is the union of the three upstream small neighborhoods adjacent to the Crossroads intersection.

Similar considerations apply to T-intersections, and we treat them in analogous manner.

### 3.3. Creating random trees

We wish to model parametrically the random tree structure of municipal sewer networks. There are two motivations for such models. The first relates to a community that may be considering our approach to homing in on COVID-19 infections. Prior to a detailed analysis of the community's sewer system, local system planners may want to obtain provisional results indicating approximate benefits (and costs) of deploying one or more sensors within the sewer system. They would like to do such an analysis without having to devote considerable time and effort obtaining and loading into a computer database the detailed map of the community's sewer system. Of course, if they already had such a detailed digital map, the modeling step would not be necessary. The second motivation for modeling is our need to test heuristic algorithms for determining the required number of sensors and their optimal or near-optimal locations.

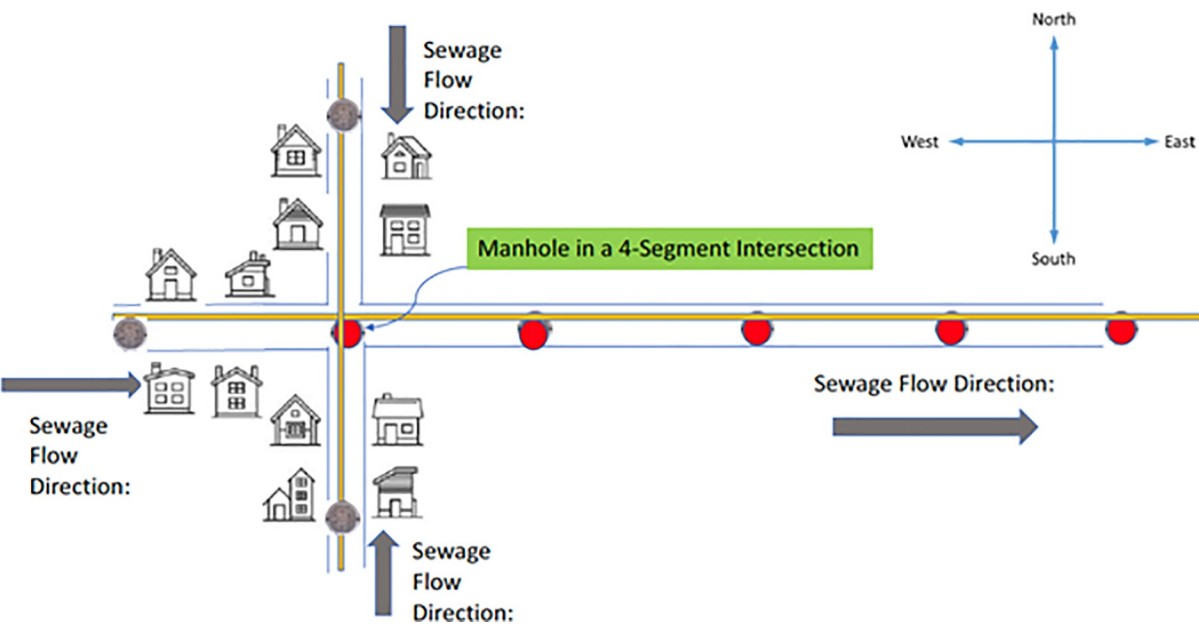

**Fig 2. Source manhole in a crossroads intersection.** Red manholes show infection; gray ones do not.

There are countless ways to create random tree graphs. We design a new process employing key attributes of the true system map. We describe these attributes probabilistically, as an empirically estimated cumulative distribution function (CDF) for street segment lengths and as an empirically estimated probability mass function (PMF) for end-of-street topologies in the simulation. Our approach is to simulate a community multiple times, then analyze results for their decision implications.

We assume that each run of the simulation has a targeted size of the resulting tree graph, where size is given as $J$, the desired number of manholes in the tree. Recall that manholes will be placed at each street (graph) intersection, at the end point of a sewer pipe dead end, and in the interior of each street segment, spaced evenly apart at a pre-specified distance, $D$. In practice, it may not be possible to terminate the tree growth precisely at the desired point $J$, but more likely at a few manholes beyond this point.

In building out the tree graph, always moving upstream, away from the WTP, we create a FIFO (First-In, First-Out) ordered list $O$ of new pipeline segments to be created. When the simulation commences, $O$ contains one pipeline segment to be created, the one emanating from the WTP. The list $O$ builds and declines over the course of the simulation. If it becomes empty during the simulation and the desired number of manholes $J$ has not been obtained, that simulation run is discarded and a new one is run to replace it.

More formally, we seek:

- The CDF $F_L(x)$ of the lengths of above-ground street segments, where $F_L(x) \equiv$ fraction of street segments less than or equal to $x$ in length. The distance units are in feet.

- The PMF $P_0(n)$ of the end-of-sewer-pipe segment tree growth outcomes, where $P_0(n) \equiv$ Fraction of end-of- sewer-pipe tree growth outcomes that are outcome $n$. There are three possible outcomes: (1) this sewer-pipe segment is a dead end, so this branch of the pipeline tree terminates any further growth of the tree from this point; (2) this sewer-pipe segment terminates under a T street intersection, so two new sewer branches will emanate from this point; (3) this sewer-pipe segment terminates under a Crossroads street intersection, so

three new sewer pipeline branches will emanate from this point. **Note**: Manholes that are to be evenly spaced in the interior of a street segment are placed there as soon as the corresponding sewer-pipe segment outcome becomes known. That is, first build the pipe under the street from street intersection to adjacent street intersection, then populate the street with evenly spaced interior manholes.

The simulation logic is shown in Box 1.

---

### Box 1. Logic of random tree generation algorithm

0. Begin at the WTP as the start node for the first street segment, the first and only entry to the set $O$. Current sum of manholes $S = 0$.

1. **Length**: From $O$, identify the next tree street segment to construct. Remove it from $O$. Obtain by Monte Carlo sampling from $F_L(x)$ the length of this new tree street segment from the given source. Once the length is obtained, deploy manholes, spaced equally, approximately $D$ feet apart over the interior of the segment. Update the sum $S$ of manholes in the tree. If $S \geq J$, **STOP**, simulation complete; else, **Continue**.

2. **End-of-street growth outcome**: Determine the tree growth outcome at the end of the just-created street segment by Monte Carlo sampling from $P_0(n)$. If $n = 1$, we terminate growth of the tree from this street segment end point and we place a manhole at the dead end; if $n = 2$ (a T-intersection), we commit to adding two new branches to the end of the current one and we place a manhole at the intersection; if $n = 3$ (a Crossroads intersection), we commit to adding three new branches to the end of the current one, and we place a manhole at the intersection. Update the sum $S$ of manholes in the tree. The new branches to add are placed at the end of the set $O$.

3. If $S \geq J$, **STOP**, simulation complete; else, **Continue**.

4. **Simulation to continue?** If we found a dead end above and there are no more new segments to create from $O$, **Terminate** this simulation and start over. Else **Continue, back to 1**.

5. **Add next street segment.** Select the first entry from $O$ and move to Step 1. Subtract this entry from $O$.

---

Returning to Belmont, Massachusetts, an eyeball inspection of the sewer pipelines indicates 12 dead ends, 18 T-intersections and only one Crossroads-intersection. This sums to 31 end-of-segment outcomes. Thus, we have $P_0(1) = \frac{12}{31} = 0.388$; $P_0(2) = \frac{18}{31} = 0.581$; $P_0(3) = \frac{1}{31} = 0.032$. Using data from the Massachusetts Department of Transportation [24], the street line segments CDF for all of Belmont are shown (in orange) in Fig 3, along with the CDF for street line segment lengths for all communities in Massachusetts (in blue). We use these data in the simulations of the next section.

## 4. The generalized tributary search algorithm

We describe our generalized "Tributary Search Algorithm" that seeks to find the source manhole. In contrast to our earlier paper [5], the generalization is required due to the more complex and realistic sewer network tree structure at street intersections, as described earlier. Here

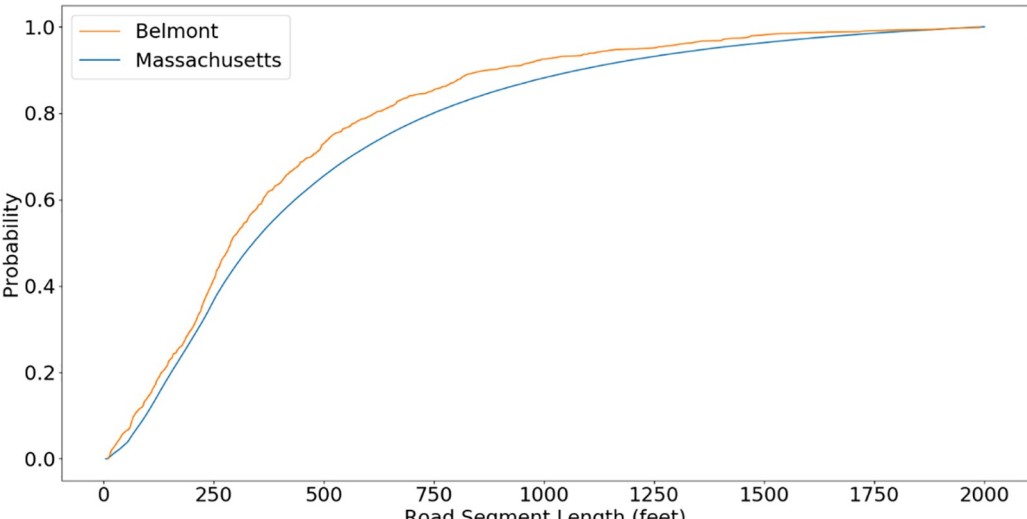

**Fig 3.** CDF for lengths of street segments for the Town of Belmont, Massachusetts (orange) and for all the communities of the Commonwealth of Massachusetts (blue).

we are applying Algorithm 1, seeking a single infected person (Newbie) in an otherwise uninfected community. Algorithm 2, seeking hot spots, has a similar logic.

Consider a directed tree network $G(M, A)$ with node set $M$ and link set $A$ in which the nodes are manholes, and the links are the sewage pipelines connecting the $|M|$ manholes. All sewage flows terminate at a single WTP node where sewage is tested and treated. Each manhole $i \in M$ has an associated Bayesian probability $\rho_i$ of being the source manhole, i.e., the closest downstream manhole to Newbie, $i = 1, 2, \ldots, |M|$. In practice, the Bayesian probabilities need to be sensibly estimated, using such inputs as the number of bedrooms in the various residences and/or neighborhood demographic information. Whenever the sewage flow of any manhole is sampled, we will know if Newbie is located upstream or downstream of the sampled manhole. If the test is positive, the infection originated upstream, and if the test is negative, the infection originated downstream.

The algorithm first computes *Bayesian probability flows* at each manhole $i$, reflecting the summed Bayesian probability that the infection is upstream of manhole $i$. Let $\delta_{ij}$ be a binary indicator equal to one if node $j$ is upstream of node $i$, and zero otherwise. The *Bayesian probability flow* entering manhole $i$ is $\sum_j \delta_{ij} \rho_j$. We seek first to sample that manhole with a *Bayesian flow probability* closest to 1/2. If the test is negative, we discard the sub-graph upstream of the tested manhole, and if the test is positive, we discard the downstream sub-graph. The discarded sub-graph, upstream or downstream, represents about 50% of the original Bayesian probabilities. We then renormalize the Bayesian probabilities and their flows in the surviving sub-graph. In that sub-graph, we again look for a node to test that has a (revised) *Bayesian flow probability* closest to 1/2. The algorithm continues in this exponentially convergent way, each time discarding a sub-graph having about 50% of the (updated) Bayesian probabilities. As described earlier, the source manhole is the most upstream manhole exhibiting remnants of coronavirus. The algorithm terminates when there is a single manhole left with an updated Bayesian probability of 1. This manhole is the Source manhole.

The algorithm generalization involves manholes at intersections, which will have two or possibly three immediately upstream manholes, one on each of the upstream intersecting streets. For an intersection manhole to be the source manhole, it must test positive *and* each of

the two (or three) immediately upstream manholes must test negative, being clear of coronavirus remnants.

We now assess the performance of the Tributary Search Algorithm. We utilize Monte Carlo simulation with random trees of various sizes, with respective numbers of manholes equal to powers of two: 16, 32, 64, 128, 256, and 512. We use the street segment CDF for Belmont Massachusetts in Fig 3 and consider an average manhole spacing of $D = 200$ feet. We consider a branching probability of $P_0(1) = 0.61$ for dead-ends, $P_0(2) = 0.28$ for T-intersections and $P_0(3) = 0.11$ for crossroads. To avoid generating trees that do not reach the required number of $|M|$ manholes, we outlaw dead ends until there are at least $|M|/4$ manholes in the tree generated thus far. To generate the Bayesian probabilities, we assign a random number, uniformly and independently generated, to each catchment zone and then normalize their sum to one. In this process, we treat crossroad intersections as having three catchment zones and T-intersections as having two catchment zones. We choose the source manhole from a Monte Carlo sampling of the Bayesian probabilities. We then apply the (Generalized) Tributary Search Algorithm and count the number of manhole samples required to find the source manhole. For each tree size, we repeat this process 30 times, each time with a different random tree. The results are shown in Fig 4 depicting the number of samples for each network size. The thickness of the boxes is proportional to the frequency of the number of samples. As an example, for 64 manholes, the probability of finding the source manhole in 6 samples is the largest. The

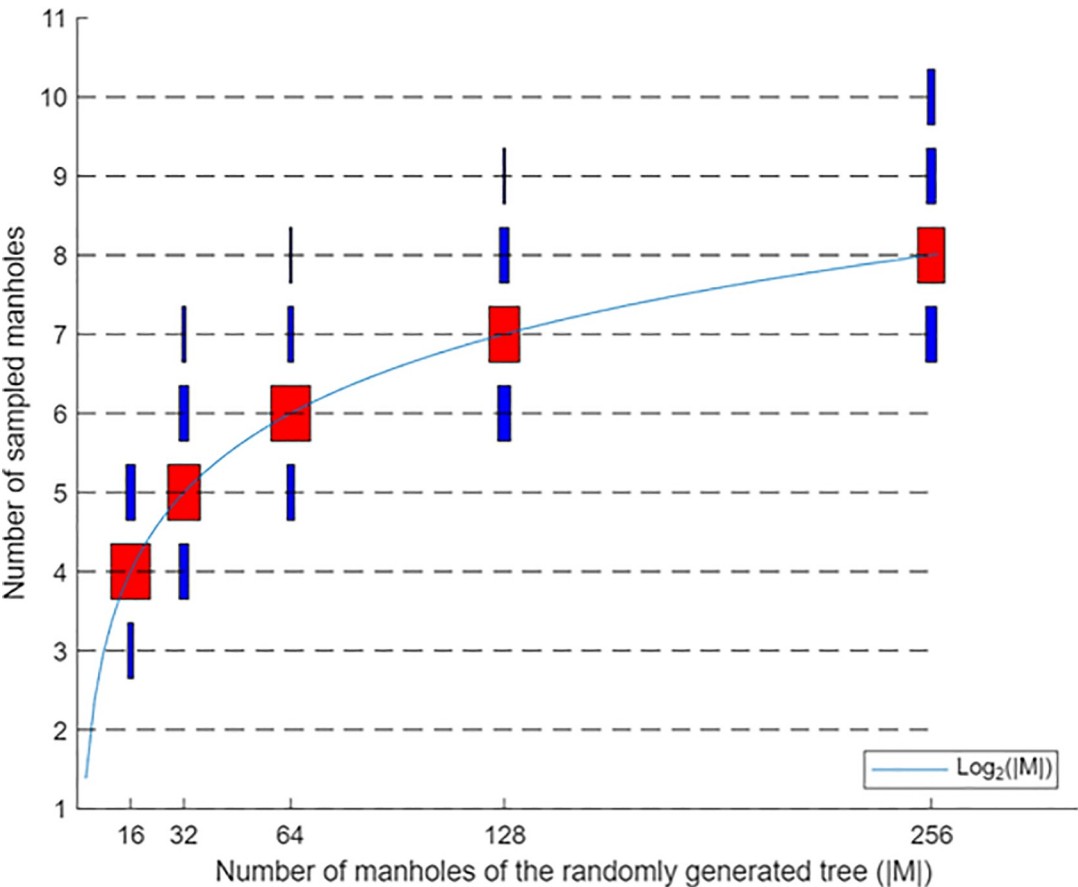

**Fig 4. Number of samples required to find the source manhole.** Additional model runs with various junction probabilities also showed the same pattern as in the figure.

figure also shows the minimum and maximum number of samples. As an example, for 64 manholes, the minimum and maximum number of samples are 5 and 8, respectively. The red boxes in Fig 4 represent the medians of the number of samples. We see that all medians are exactly equal to $\log_2 |M|$.

The 50% cut at each iteration of the Tributary Search Algorithm seems to suggest an exponentially convergent process. According to Fig 4, the average number of samples of our Tributary Search Algorithm is similar to the number of iterations required of the binary search algorithm, which is $\log_2 |M|$ for the case of equal Bayesian probabilities in a straight-line network with $|M|$ consecutive manholes [25].

This is an important finding because it allows us to use a closed-form expression of the number of samples for a given network with $|M|$ manholes. This expression can be used to find the required number of sensors in the network if one wants to keep the number of samples below a threshold and ultimately enables formulating the sensor location problem as a mathematical model for which we can develop solution algorithms. It also allows analyzing the structural properties of the sensor location problem including robustness to perturbations in sensor locations. Moving forward, for notational convenience, we omit the number (base) 2 in $\log_2()$.

As mentioned earlier, in Larson et al. [5] we provide two algorithms: one for finding the single source of contamination and one for finding the hot spot (the neighborhood with the highest level of contamination). It was shown in Larson et al. [5] that the second algorithm (finding the hot spot) is a simple extension of the first and the approximation of Fig 4 still holds.

## 5. System sizing and the sensor location problem

The two objectives of our overall problem are to find the appropriate number of sensors and their respective locations in the wastewater sewer network.

### 5.1 System sizing

Avoiding further spreading of the virus requires finding the source manhole quickly, in as few manhole samples as possible, preferably in the course of one day. The samples cannot be taken in parallel because the result of one sample guides to the location of the next sample to be taken; the inherent dependencies between consecutive samples elongate the process. We will require that a maximum expected value of $T$ samples is allowed for avoiding unacceptable delays. Usually, if $T \leq 8$, convergence to the source manhole can occur in one working day. While it is intuitive that more sensors will reduce the number of manhole samples, the explicit relationship between the number of sensors and the number of manhole openings requires some analysis.

We first introduce the concept of an *entry set* which is fundamental to the description of the problem. A manhole is included in the entry set of a sensor if that sensor is the first to detect the presence of virus originating from that manhole. Analogous to catchment zones, the entry sets are mutually exclusive and collectively exhaustive. The WTP, always equipped with virus detection technology, has its own entry set. Thus, if our algorithms install $S$ sensors, there are a total of $S+1$ entry sets in the system.

We now develop a "back-of-the-envelope" approximation for the performance of the system as a function of the number of sensors deployed. Suppose $S$ sensors are located in a wastewater network with $|M|$ manholes, each with identical Bayesian probability, $1/|M|$. Let $Q(S)$ be the expected number of manhole samples required to find the source manhole. Suppose an equal number of manholes $|M|/(S+1)$ are allocated to each entry set. Therefore, the expected

number of samples can be written as

$$Q(S) = \log \frac{|M|}{S+1}. \tag{1}$$

Recalling that a maximum of $T$ samples is allowed, we require $\log \frac{|M|}{(S+1)} \leq T$. System planners would prefer to have the smallest possible number of sensors, $S^*$, so that now we require $\lceil \log \frac{|M|}{(S^*+1)} \rceil = T$. From this, we have a fundamental relationship,

$$S^* = \lceil \frac{|M|}{2^T} - 1 \rceil. \tag{2}$$

We see that $S^*$ increases linearly with size of the network as measured by $|M|$, and decreases exponentially with the allowable number of samples $T$. Fig 5 shows this relationship. As an example, for 2,000 manholes and $T = 8$ manual samples, we need $S^* = \lceil \frac{2000}{2^8} - 1 \rceil = 7$ sensors.

There are sharp diminishing returns to scale in the impact of sensors. By adding $S$ sensors to a system having no sensors, from (1) we reduce the expected number of manhole samples by $\Delta = Q(0) - Q(S) = \log|M| - \log \frac{|M|}{(S+1)} = \log(S + 1)$. Accordingly, if we deploy one sensor, i.e., $S = 1$, then we reduce the expected number of manhole samples by one, as the sensor acts on behalf of an otherwise required first manhole sample. If we plan to reduce the expected number of manhole samples again by one (total by two), we would need two additional sensors, one located within the entry set of the first sensor and the other located in the entry set of the WTP. The number of required sensors to reduce the expected number of manholes samples by $\Delta$ is $S = 2^\Delta - 1$, which is an exponential function, demonstrating that the first few installed sensors have a much greater impact on the expected number of manhole samplings than the latter ones.

We note that the number of sensors given in (2) is based on two strong assumptions. First, it assumes all manholes have the same Bayesian probability, a useful assumption for obtaining an approximate number of sensors but may require fine-tuning when the Bayesian

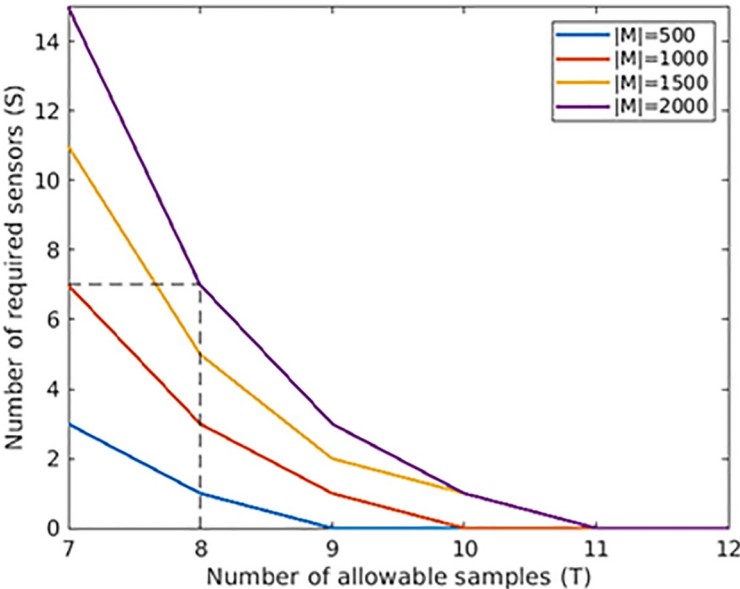

**Fig 5. Number of required sensors given the allowable number of manhole samples.**

probabilities are not equal. Second, it assumes that the *expected number* of manholes can be used, ignoring the probability distribution.

## 5.2 The sensor location problem

We seek to find the locations of the $S$ sensors in the wastewater network that lead to the best (smallest) expected number of manhole samples. Let $x_i$ be a binary variable equal to one if a sensor is located at manhole $i$, and zero otherwise. Given a set of $S$ locations for the sensors, let $m_i$ be an integer decision variable representing the number of manholes contained in the entry set of the sensor located at node $i$. The following non-linear integer program finds the best set of sensor locations.

$$\text{Min. } \sum_i \log(m_i) p_i x_i \tag{3}$$

$$m_i = \sum_j \delta_{ij} - \sum_j \delta_{ij} m_j x_j \qquad \forall i \tag{4}$$

$$p_i = \sum_j \delta_{ij} \rho_j - \sum_j \delta_{ij} p_j x_j \qquad \forall i \tag{5}$$

$$\sum_i x_i = S \tag{6}$$

$$x_i = \{0, 1\}, m_j \in Z \qquad \forall i \tag{7}$$

Given $S$ sensors to locate, using the $\log(|M|)$ approximation, the minimization of the objective function in (3) seeks the smallest expected number of manhole samples required to find the source manhole. Constraint (4) gives the number of manholes in the entry set of sensor $i$, $\forall i$. Constraint (5) gives $p_i$, the sum of Bayesian probabilities of the manholes in the entry set of sensor $i$, $\forall i$. Constraint (6) ensures that $S$ sensors are located, and Constraint (7) identifies the decision variables. The mathematical model is a non-linear integer program. The non-linearity arises from the log function in the objection function and the product of terms in both the objective function and the Constraints (4) and (5).

The problem cannot be solved efficiently using existing solvers because the formulation is a non-linear integer program with a huge number of decision variables: $2|M|$ integer and binary decision variables and $2|M|+1$ constraints (recall that typical $|M|$ is between 250–5000). We later show that the number of samples is insensitive to the exact location of the sensors. This justifies our use of a heuristic algorithm explained in the next section. In the following heuristic algorithm, we set $0 \times \log 0 = 0$ in the objective function (3), implying that if a manhole $i$ does not have a sensor the term $\log(m_i) p_i x_i$ does not appear in the objective function. This convention allows us to circumvent the computational complexities of dealing with $\log(0)$ which has a right-side limit of negative infinity.

## 5.3 A heuristic algorithm

We develop a decomposition heuristic to solve the above non-linear integer program. Definition: A sensor $s_b$ is defined to be a sensor located "immediately upstream" of sensor $s_a$ if wastewater from $s_b$ flows next into one of the manholes in the entry set of sensor $s_a$. Note that a sensor can have multiple immediately upstream sensors. Using the same convention, if $s_b$ is immediately upstream of $s_a$ then $s_a$ is immediately downstream of $s_b$. Note that each sensor has only a single immediately downstream sensor due to the tree structure of the network. When there is only one sensor in the network, its immediately downstream sensor is the WTP.

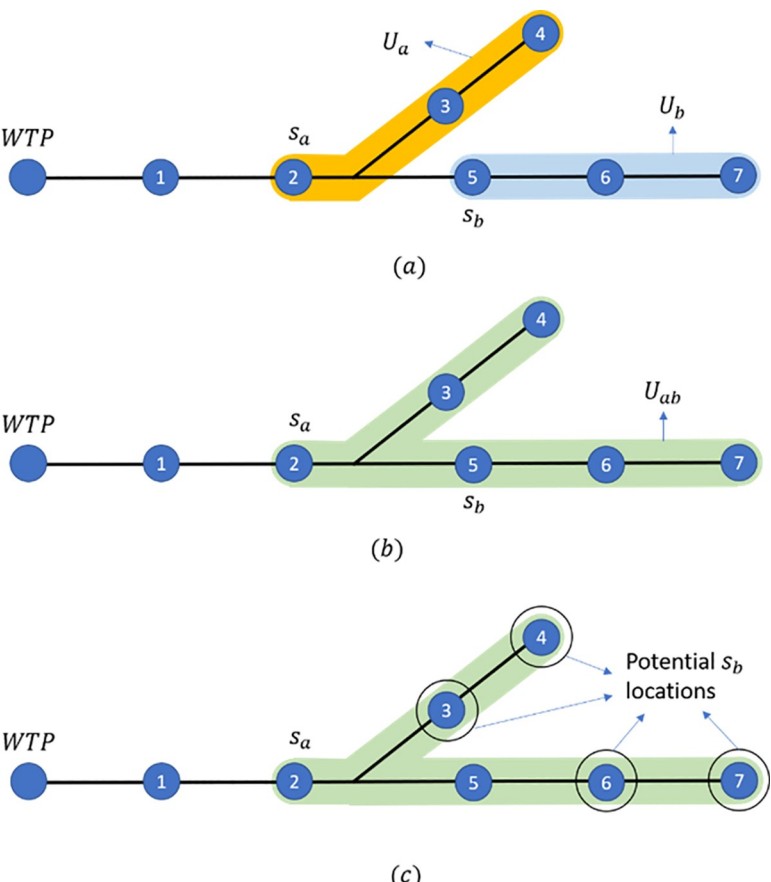

**Fig 6.** a) entry sets of the two sensors, b) the union of the entry sets, and c) potential locations for the immediate upstream sensor.

The heuristic requires several additional definitions. Let $(s_a, s_b)$ be a paired set of two sensors such that $s_b$ is immediately upstream of $s_a$. For $S$ sensors there are always $S+1$ such combinations because the WTP also has its own entry set. Note that the WTP can only be a downstream sensor.

Let $U_i$ be the set of manholes in the entry set of sensor $s_i$ and let $U_{ab} = U_a \cup U_b$ be the union of the sets of manholes in the entry sets of $s_a$ and an immediately upstream sensor $s_b$, as shown in Fig 6A and 6B. In the Algorithm below we will fix the location of $s_a$ and consider every potential $s_b$ in $U_{ab}$ as shown in Fig 6C. Note that $s_b$ always remains in the upstream of $s_a$. The algorithm for the sensor location problem includes the following steps:

> ## Box 2. Algorithm for the sensor location problem
>
> 1. **Initialize:** Choose initial locations of $S$ sensors. Initialize the empty set $V = \emptyset$ as the set of already evaluated pairwise sensor combinations.
>
> 2. Find all **pairwise combinations** of sensors $(s_a, s_b) \notin V$ such that $s_b$ is immediately upstream of $s_a$. Compute $Q_{ab} = \log(m_a)p_a + \log(m_b)p_b$.
>
> 3. **Select** the largest $Q_{ab}$. In the case of ties, choose arbitrarily.

4. **Find** the location of $s_b$ in $U_{ab}-\{s_a\}$ that minimizes $Q_{ab}$. In a case of a tie choose arbitrarily. Fix the location of $s_b$ and update $Q_{ab}$. If there is a reduction in $Q_{ab}$, set $V = \{(s_a, s_b)\}$, otherwise set $V := V \cup \{(s_a, s_b)\}$.

5. **Terminate** if set $V$ contains all the pairwise combinations. Otherwise, go to Step 2.

In Step 1 of the algorithm, a good initial solution is to locate the sensors so that the entry sets hold approximately equal number of manholes. For instance, if there is one sensor and nine manholes, one entry set would contain four manholes and the other five. We show in Lemma 1 (in S1 Appendix) and Result 1 below that this solution is optimal if the Bayesian probabilities are all equal. In Steps 2 and 3 of the algorithm we select the largest $Q_{ab}$ not yet considered because this term has the highest potential influence on the objective function value. In Step 4, the algorithm seeks to reduce this term in the objective function by moving the one upstream sensor to other possible locations in the union of the two respective entry sets. If a net reduction is found, sensor $s_b$ is moved to its new (better) location, and potentially all system entry sets are redefined. In that case, all current entries to the stack $V$ are erased and $V$ is now populated only with the new pair $(s_a, s_b)$. All other pairs, even if previously included in $V$, will need to be re-analyzed due to the system entry-set reconfiguration caused by relocating $s_b$. If we cannot improve the solution by moving $s_b$ within the union set $U_{ab}$, then the sensor pair $(s_a, s_b)$ is added to current set $V$. When all sensor pairs are considered and the solution cannot get any better, we terminate this steepest-descent heuristic algorithm as set $V$ will include all pairwise sensor combinations.

Consider the initial solution explained above, where we locate sensors so the entry sets are roughly equal in number of manholes.

**Results 1.** In a network with $S$ sensors, $|M|$ manholes, and equal Bayesian probabilities of $\rho = 1/|M|$, to minimize the number of samples, the sensors are located to have approximately equal number of manholes in their respective entry sets.

Result 1 is based on Lemma 1 in the S1 Appendix. In the proof of Lemma 1 given in the S1 Appendix we show that any perturbation to this solution would increase the expected number of manhole samples.

In the S1 Appendix, we also present Lemma 2 and its proof that shows a special case of a wastewater tree with no junctures for which Step 4 of the algorithm can be accelerated to avoid checking every manhole in $U_{ab}$. Whenever the union set $U_{ab}$ does not include any junction, we can use Lemma 2 and replace step 4 of the algorithm with the following alternative 4' for which the steepest descent approach is used to find the best location of sensor $s_b$ in fewer algorithmic checks. Step 4': First, locate $s_b$ in a manhole that has an equal number of manholes in its $U_{ab}$ upstream and downstream or, if the sum of upstream and downstream manholes is an odd number, locate $s_b$ so that the difference between the numbers of $U_{ab}$ upstream and downstream manholes is one. Check the two manholes adjacent to $s_b$ and relocate the sensor to the manhole with the lower $Q_{ab}$. Continue until the two manholes adjacent to $s_b$ have a larger $Q_{ab}$.

## 5.4 Simple example

Consider the simple example of Fig 7. We index manholes by numbers and sensors by letters. All manholes have a Bayesian probability of 0.1 except for manhole 4 which has a Bayesian probability of 0.3. Without loss of generality, we assume in this example that the WTP ($s_w$) also has an entry set with a Bayesian probability of 0.1 (i.e., the WTP directly collects wastewater

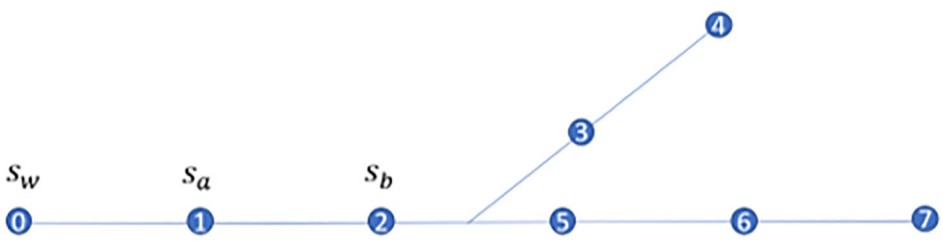

a- Initial solution

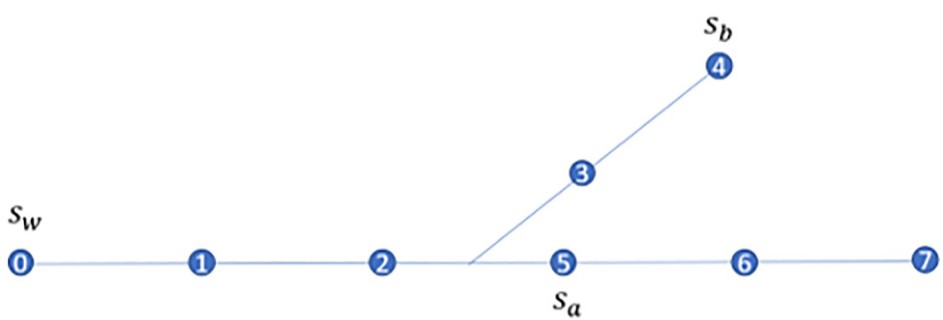

b- Ending solution

**Fig 7. Initial and best solution of the sensor location problem example.**

from nearby residences). There are two new sensors in the network represented by $s_a$ located at node 1 and $s_b$, located at node 2 as depicted in Fig 7A.

The entry set of the WTP located at $s_w$ contains only manhole 0, the entry set of sensor $s_a$ contains manhole 1, and the entry set of sensor $s_b$ contains manholes 2–7.

There are two sensor combinations: $(s_w, s_a)$ and $(s_a, s_b)$. We calculate $Q_{wa}$ and $Q_{ab}$ associated with the two pairs in the objective function: $Q_{wa} = \log(1)\times0.1+\log(1)\times0.1 = 0$ and $Q_{ab} = \log(1)\times0.1+\log(6)\times0.8 = 2.06$. Because $Q_{wa}<Q_{ab}$, we choose the pair $(s_a, s_b)$ and set $V = \{(s_a, s_b)\}$, keep $s_a$ at node 1, and relocate the sensor $s_b$ within the joint entry set $U_{ab}-\{s_a\}$ which includes

**Table 1. Iterations 1 and 2 of the heuristic.**

| Log(2) Iteration 1 | | | | | | |
|---|---|---|---|---|---|---|
| Sensor $s_b$ location (manholes number) | 2 | 3 | 4 | 5 | 6 | 7 |
| $Q_{ab}$ | 2.06 | 1.56 | 1.56 | 1.66 | 1.83 | 2.06 |
| Iteration 2 | | | | | | |
| Sensor $s_a$ location (manholes number) | 1 | 2 | 3 | 5 | 6 | 7 |
| $Q_{wa}$ | 1.56 | 1.36 | 1.56 | 1.27 | 1.36 | 1.56 |

manholes 2 to 7. In the first iteration in Table 1 we show $Q_{ab}$ for every $s_b$ in $U_{ab}$. For example, when $s_b$ is moved to node 3 then $Q_{ab} = \log(5)\times0.5+\log(2)\times0.4 = 0.56$.

Since there is a tie in the minimum $Q_{ab}$, we arbitrarily choose to relocate sensor $s_b$ to manhole 4 (instead of manhole 3). Because we find a smaller $Q_{ab}$ we fix the location of $s_b$ at node 4 and set $V = \{s_a, s_b\}$.

The network has only one sensor pair $(s_w, s_a)$ not included in $V$. We calculate $Q_{wa} = 0.6 \log(6)+0.3 \log(1) = 1.56$, and move the location of sensor $s_a$ (immediately upstream of $s_w$) within the joint entry set of $U_{wa}-\{s_w\}$, which includes manholes 1–3 and 5–7. The revised $Q_{wa}$ is presented in Iteration 2 of Table 1. We choose to locate the sensor $s_a$ to manhole 5 which gives the smallest $Q_{wa} = 1.27$. We now reset the set $V:=\{(s_w, s_a)\}$. If we continue the algorithm for one more iteration, the solution will not improve, set $V$ will be updated to $V:=\{(s_w, s_a), (s_a, s_b)\}$, and the algorithm terminates.

Because of the small size of the network in Fig 7, we were able to enumerate exhaustively all the solutions and find the optimal solution by brute force. Doing that, the optimal solution is to locate one sensor on manhole 4 and the other on manhole 5 to reach the lowest objective function value of $0.4 \log(4)+0.3 \log(1)+\log(3) 0.3 = 1.27$. In this example, the optimal solution is the same as the heuristic solution.

## 5.5 Location algorithm robustness

The presented location algorithm is a heuristic that may not find the optimal location of the sensors. Nevertheless, given the special structure of the problem and the properties of the log function in the objective function, the solution of the heuristic is robust in that perturbing it does not significantly change the objective function value. The presented location problem is insensitive to such minor perturbations, yielding some practical advantages. First, in some wastewater networks the sensors might be best installed in pumping stations for practical reasons. If the pumping stations are close to the sensors' locations obtained from the heuristic, we can simply move the sensors from the manholes to the nearest pumping stations without jeopardizing the quality of the solution. Second, the Bayesian probabilities do not need to be very accurate. As an example of these two points, consider the simple network in Section 5.4. As mentioned earlier for this example all possible solutions can be obtained by brute force. It is easy to verify that the values of the objective function of the top one-third of the of solutions are within 10% of the optimal solution.

Assume the previous setup as above where $S$ sensors (including the WTP) are optimally located in a network with $|M|$ manholes each with Bayesian probability $\frac{1}{|M|}$. We show in the proof of Lemma 1 that if we move one sensor to an adjacent manhole, then the objective changes from $\log \frac{|M|}{S+1}$ to

$$\frac{s-1}{s+1}\log\left(\frac{|M|}{S+1}\right) + \left(\frac{1}{s+1} + \frac{1}{|M|}\right)\log\left(\frac{|M|}{S+1}+1\right) + \left(\frac{1}{s+1} - \frac{1}{|M|}\right)\log\left(\frac{|M|}{S+1}-1\right).$$

The error ratio, $\epsilon$, is

$$\epsilon = \frac{\frac{s-1}{s+1}\log\left(\frac{|M|}{S+1}\right) + \left(\frac{1}{s+1} + \frac{1}{|M|}\right)\log\left(\frac{|M|}{S+1}+1\right) + \left(\frac{1}{s+1} - 1\frac{1}{|M|}\right)\log\left(\frac{|M|}{S+1}-1\right) - \log\frac{|M|}{S+1}}{\log\frac{|M|}{S+1}}. \tag{8}$$

If we vary the number of sensors from 1 to 12 and the number of network manholes from 500 to 2,000, the largest error ratio is an infinitesimal $1.5\times10^{-5}$. This optimal location problem is indeed extremely insensitive to exact locations.

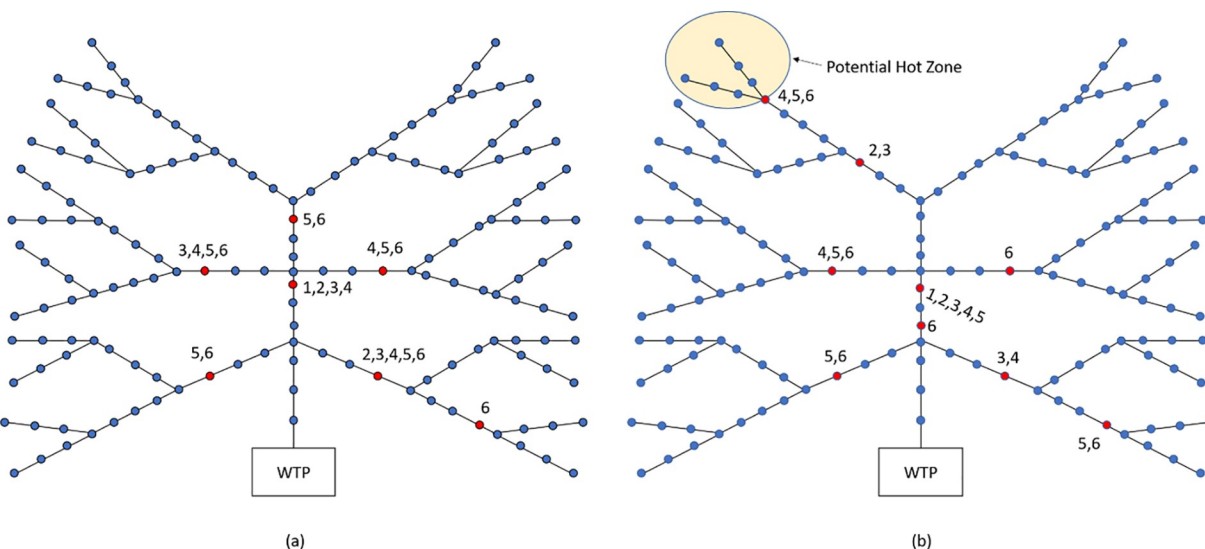

**Fig 8. Optimal locations of 1 to 6 sensors.** The number # on the red nodes indicates that the manhole was chosen when locating # sensors. Panel b shows the hotspot.

## 6. Numerical illustration

We present another numerical example to gain more insights on the number of sensors and their locations in the network. We first assume that the Bayesian probabilities are randomly generated from a unit uniform distribution normalized so that they sum to one. Fig 8A presents the best locations of one to six sensors. If we have only one sensor it is best to locate it such that approximately half of the manholes are upstream and the other half downstream of the sensor. As we increase the number of sensors, they become more spread out in the network to reduce the number of manhole samples required. Moving to Fig 8B, we perform the same test, this time requiring that there is a 30% probability the source manhole is contained in the hotspot shown. Fig 8B also shows the optimal location of one to six sensors. If we install more than three sensors, then at least one sensor is adjacent to the hotspot as depicted in Fig 8B. The spreading of the sensors (Fig 8B) is also more skewed towards the hotspot compared to the case of uniform Bayesian probabilities (Fig 8A).

We next illustrate the stochasticity in the number of manhole samples. Consider a randomly generated network with 2,000 manholes created as explained in Section 2. The Bayesian probabilities are uniformly distributed. For each scenario $S$ we Monte Carlo choose the location of the source manhole 30 times according to the Bayesian probabilities. Fig 9A shows the manhole sample distribution at various number of sensors. The average number of samples decreases with the number of sensors. Adding more sensors reduces the average and/or the variance of the number of manhole samples. For a confidence level of $\alpha = 0.5$, and an acceptable number of samples, $T = 7$, we need at least 8 sensors. However, if we increase the confidence level to $\alpha = 0.8$ we need more than 9 sensors.

Suppose the sensors can only be installed on designated stations spread out in the network. To assess the impact of this modification, we first solve the original location problem without any restriction (as shown in Fig 9A) to find the best location of the sensors. We next move every sensor to the nearest pumping station. Because there are more stations than sensors, we rarely experience the case where two or more sensors have the same closest station. In such scenarios, we use a simple assignment heuristic where we first sort the pair-wise combinations of sensors and stations according to their distance (i.e., the number of manholes between the

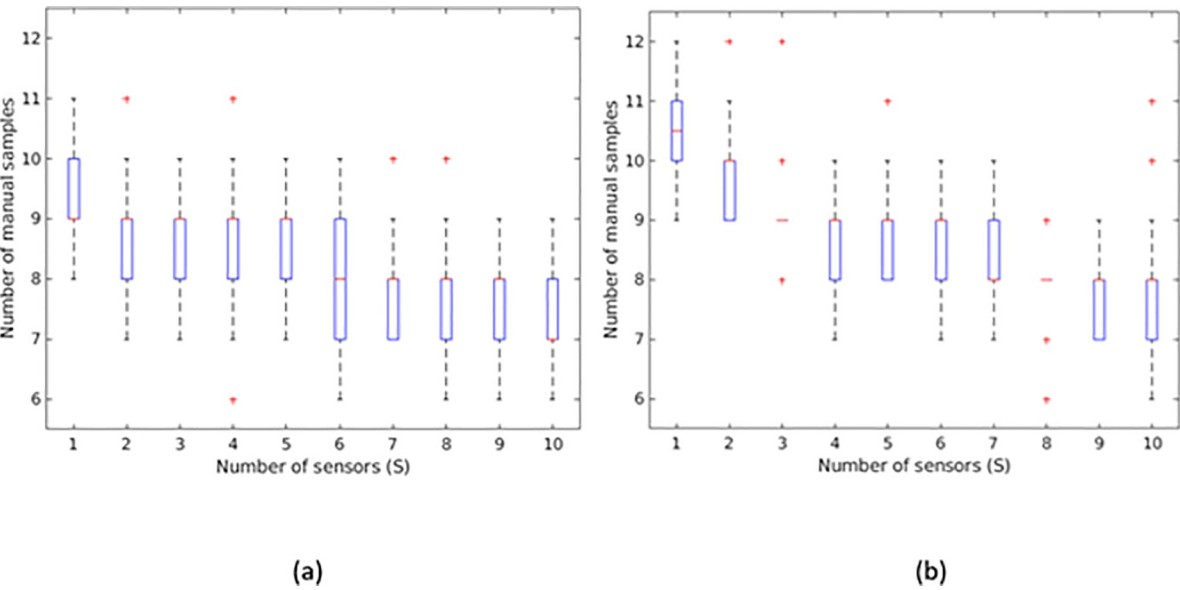

**Fig 9. Number of manhole samples when S sensors are installed when.** a) there are no restrictions on the location of the sensors, and b) when sensors can only be installed at pumping stations. The red crosses are outliers.

sensor and the station). We assign the first item on the list, the pair-wise combination with the smallest distance and discard all other combinations that include the assigned sensor or station. We continue the process until all sensors are assigned to a pumping station. We use the aforementioned network with 2,000 manholes and uniformly generated Bayesian probabilities and assume there are 500 stations where the sensors can be installed. The number of manhole samples is presented in Fig 9B. Statistical t-tests with confidence level 0.95 show there is no significant difference between the results in Fig 9A and 9B.

# 7. Reflections on needed additional R&D and extended domains of applicability

## 7.1 Required R&D

Certain research-and-development (R&D) efforts are needed for professionals to be able implement our proposed sewer monitoring system in practice. In that sense, our work can be viewed as a motivational blueprint for the future, with investments in water and sewer sensor-based systems projected to grow by billions of dollars in the next ten years. And while the present 2020–2021 waves of Coronavirus are subsiding, due to vaccines and herd immunity, there are projections that the virus will continually mutate and return year after year, not unlike seasonal influenza. If that were to occur, our proposed system should prove useful indefinitely into the future.

We see the need for three major R&D efforts. The first is to be devoted to creation of a fast, inexpensive test that can be done by non-experts sampling sewage at manholes. As discussed in our previous paper [5], serious research is being done now on developing such tests. Already, our modeling and algorithmic work has revealed the strong need for such tests, thereby spurring new development. Dr. Zhugen Yang at (Cranfield Water Science Institute, Cranfield University) and colleagues have been working on this problem for over five years [26, 27]. At present, we are collaborating with him on his quest to develop an accurate and

inexpensive paper-based test for detecting remnants of Coronavirus as well as other pathogens in raw sewage. We are hopeful that such tests may become available in 2021 or 2022 [28].

The second major R&D need is development and testing of in-manhole sensors that can rapidly signal the presence of Covid-19 genetic remnants. As discussed above, there are currently available in-system sensors to detect various chemicals and other unwanted intrusions into sewage systems. But current detection of Covid-19 genetic materials usually requires removal of the sample from the manhole and doing the test in a laboratory. As of this writing, significant effort is in Israel, where Ben-Gurion University of the Negev has signed with the Health Ministry to monitor and detect COVID-19 in sewage samples from 14 Israeli communities. Ben-Gurion researchers are collaborating with two Israeli companies, NUFiltration and Kando. According to media accounts, "NUFiltration is helping to develop a filtration device to detect concentrations of coronavirus in wastewater, and Kando installs a kit of sensors and smart samples in manholes of sewer networks to identify where the highest concentrations of COVID-19 are located" [29]. This work, as described, is very close to the type of in-manhole sensor system described in this paper [30].

The third major area for needed R&D is the effect of sewer system hydraulics on the efficacy of our proposed Coronavirus detection and search method. The signaling of Coronavirus at a sensor or the WTP may be received minutes or even hours after a toilet was flushed containing the infection marker. Once detected at a sensor or the WTP, a directed sequence of manhole openings and sewage testing is initiated. Assuming the availability of a fast on-scene test, that manhole testing can take up to one working day to complete. The key issue is whether the Coronavirus signal remains detectable in the sludge of manholes through which it passed or if it dies off prior to testing. We've heard conflicting opinions by experts. Only field testing will answer this question. Regarding sewer systems and their hydraulic properties, some systems–especially newer ones, could support the type of system we propose. But numerous sewer systems in the U.S., including in New England, are many decades old, with portions crumbling and wastewater being mixed with storm-drain water [31, 32]. For such dilapidated systems, we doubt that our proposed methods would work. But we are optimistic that the proposed system would work well in newer and "tight" sewer systems. Additional R&D is required to test our optimism.

## 7.2 Extended domains of applicability

Regarding the applicability of our methods to other applications, we see many opportunities. To our knowledge, we are the first to suggest binary search techniques for selecting manholes to open. The binary search exponentially reduces the number of candidates to test; for a search region having 250 manholes, for instance, the average number of required tests to identify the source manhole is 8. This gives the practitioner searching for an address (in our case, of a person newly infected with Coronavirus) a degree of specificity not available just by knowing the location of the first sensor to broadcast an alarm. Virtually all of the sensor-location papers we reviewed for Section 2.3, in both sewage and water systems applications, ended with the address of the first triggered sensor. No additional address information was available or sought. Our binary search methods applied to sewage systems, opening and testing manholes intelligently via binary search, yield a locational resolution typically two orders of magnitude better than the sensor-only approach. For the case of water distribution systems, we would need to identify in-system testing points analogous to manholes in sewage systems; this should not be difficult to do.

Searching for addresses via sewage networks does not have to be restricted to persons newly infected with Coronavirus. Other diseases such as polio have been tracked via sewage flow

analysis [33]. As WBE has matured in recent years, researchers have been able to track in sewage Hepatitis A Virus and Norovirus (in Sweden) [34] and diarrhea-causing noroviruses (in Italy) [35]. For all of these diseases, the location resolution available from our binary search methods is significantly better than only knowing the address of a sensor, or in the case of no sensors, only the address of the wastewater treatment plant! But a word of caution, all the R&D and implementation issues addressed above must be solved for each of these diseases for the binary search to be used successfully in practice.

Searching through sewer systems does not have to be restricted to human diseases. With ever-improving methods of WBE, one can detect in sewage the presence illicit opioid drugs such as fentanyl, heroin, oxycodone, and codeine [36]. Such detection usually implies the human ingestion of these drugs, a national tragedy that by overdosing kills over 50,000 Americans each year. As one final application, via sewer searching, one can locate locations of bomb makers and chemists with crystal-meth labs [37]. Indeed, a community's sewer system contains almost limitless information on human health and behavior.

## 8. Summary

We have taken a mathematical modeling approach to an urgent societal problem: early identification of individuals newly infected with Coronavirus, the intent being to remove them from the general population early in order to minimize chances of them infecting others and to get them the professional medical care they need. Such individuals may be pre-symptomatic or asymptomatic, not even knowing they are infected, but perhaps highly infectious to others. Using sewer flows and wastewater-based epidemiology (WBE), one can detect a new infection up to one week prior to physical symptoms of illness appearing. Effective use of wastewater virus signals can significantly reduce the spreading of infection, most likely saving lives.

In this paper, we have extended our previously reported Tributary Search Algorithm [5] for sequentially opening and testing manholes and proposing a new technology–in-sewer sensors–to act as 24/7 sentinels for virus signals. Adding sensors to the system speeds the finding of Newbie and/or speeds the finding of hot zones of infection. The proposed in-sewer sensor system signifies a type of targeted *pooled sampling* for communities of many thousands of residents.

Following traditional modeling analysis steps of problem framing, formulation, analysis and solution, we devised easy-to-compute procedures for answering the two key questions with sensors: How many? And where to locate them? The analysis employed both Monte Carlo simulations and optimization models. Using simulation with Massachusetts data, we verified the accuracy of the $log_2()$ function in computing the mean and median number of manhole openings for a problem of given size. From that result, we were able to develop robust methods for system sizing (i.e., finding the best number of sensors) and heuristic methods for best locating a given number of sensors.

A major finding is the significant insensitivity of results to exact locations of the sensors. With equal Bayesian probabilities on sewage flows, our rule-of-thumb guidance is to design respective sensor entry sets so that each has approximately the same number of manholes. This guidance is adjusted for non-equal Bayesian probabilities, but the insensitivities to exact locations remain. We hypothesize that many of the water system and sewage system references cited in Subsection 2.3 might also have benefited from an analysis of system performance insensitivity to exact locations. The persistent focus on obtaining "optimal locations" might have been misplaced.

We very much hope that this work will lead to follow-on efforts by professionals with the necessary engineering and WBE skills, hopefully heading to design and manufacture of sensors of various types and successfully placing them into modern operating sewer systems.

## Supporting information

**S1 Appendix. Proofs of Lemmas 1 and 2.**
(DOCX)

## Acknowledgments

We sincerely thank Evan Larson for his most helpful work in locating sewer system maps and utilizing roads data from the Massachusetts Department of Transportation to create Fig 4. For his supportive and open discussions relating to the possible implementation of these methods, we thank Marc F. Valenti, CPWP-S, Manager of Operations, Dept. of Public Works, Town of Lexington, Massachusetts. And for his detailed review of an earlier draft of this paper and for his on-going encouragement and advice in this active area of research, we thank Professor Ed Kaplan of Yale University.

## Author Contributions

**Conceptualization:** Mehdi Nourinejad, Oded Berman, Richard C. Larson.

**Investigation:** Mehdi Nourinejad, Oded Berman.

**Methodology:** Mehdi Nourinejad, Oded Berman, Richard C. Larson.

**Software:** Mehdi Nourinejad.

**Validation:** Mehdi Nourinejad, Richard C. Larson.

**Visualization:** Mehdi Nourinejad, Oded Berman, Richard C. Larson.

**Writing – original draft:** Mehdi Nourinejad, Oded Berman, Richard C. Larson.

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
