## [Decision Letter · Decision Letter 0]

27 Jan 2021

PONE-D-20-38252

Placing Sensors in Sewer Networks A System to Pinpoint New Cases of Coronavirus

PLOS ONE

Dear Dr. Nourinejad,

Thank you for submitting your manuscript to PLOS ONE. After careful consideration, we feel that it has merit but does not fully meet PLOS ONE’s publication criteria as it currently stands. Therefore, we invite you to submit a revised version of the manuscript that addresses the points raised during the review process.

We look forward to receiving your revised manuscript.

Kind regards,

Gabriele Oliva, Ph.D

Academic Editor

PLOS ONE

Additional Editor Comments:

Two reviews were collected. Although both reviewers acknowledge the interesting nature of this work, both raise concerns that should be addressed and both suggest major revision. After carefully checking the manuscript myself I agree with the reviewers' evaluation and I recommend a major revision.

Journal Requirements:

4. We note that Figure 1 in your submission contain map images which may be copyrighted. All PLOS content is published under the Creative Commons Attribution License (CC BY 4.0), which means that the manuscript, images, and Supporting Information files will be freely available online, and any third party is permitted to access, download, copy, distribute, and use these materials in any way, even commercially, with proper attribution. For these reasons, we cannot publish previously copyrighted maps or satellite images created using proprietary data, such as Google software (Google Maps, Street View, and Earth). For more information, see our copyright guidelines: http://journals.plos.org/plosone/s/licenses-and-copyright.

4.1.    You may seek permission from the original copyright holder of Figure 1 to publish the content specifically under the CC BY 4.0 license. 

4.2.    If you are unable to obtain permission from the original copyright holder to publish these figures under the CC BY 4.0 license or if the copyright holder’s requirements are incompatible with the CC BY 4.0 license, please either i) remove the figure or ii) supply a replacement figure that complies with the CC BY 4.0 license. Please check copyright information on all replacement figures and update the figure caption with source information. If applicable, please specify in the figure caption text when a figure is similar but not identical to the original image and is therefore for illustrative purposes only.

Reviewers' comments:

Reviewer's Responses to Questions

**Comments to the Author**

1. Is the manuscript technically sound, and do the data support the conclusions?

Reviewer #1: Partly

Reviewer #2: Partly

2. Has the statistical analysis been performed appropriately and rigorously? 

Reviewer #1: N/A

Reviewer #2: N/A

3. Have the authors made all data underlying the findings in their manuscript fully available?

Reviewer #1: Yes

Reviewer #2: Yes

4. Is the manuscript presented in an intelligible fashion and written in standard English?

Reviewer #1: No

Reviewer #2: No

5. Review Comments to the Author

Reviewer #1: Dear Editor

The paper deals with a method for solving sensor placement problem in a sewer system. The Authors formulate the sensor location problem as an integer nonlinear optimization and develop heuristics to solve it. The topic is of high interest for the scientific community. However, the paper often is confused, redundant and wordy (there is possibility to save space for better discussing the application of the proposed framework) and a vigorous revision of the English language is necessary. Furthermore, the paper is not presented in an appropriate fashion (especially the state of art and the discussion of results) and some crucial aspects should be clarified (in particular the methodology). I recommend major revisions be undertaken; listed are comments and suggestions I would like to see addressed before the acceptance of the paper:

The paper lacks a proper state of art about the importance of the monitoring of the water systems, in general, and more specifically of sewer systems and methods already developed for addressing the sensor placement problem, as well as, about the study of the topology of sewer systems. I strongly suggest to improve the Introduction. In this regard, I list some papers could be of your interest to refer to:

- Banda, T. D., & Kumarasamy, M. (2020). Development of a Universal Water Quality Index (UWQI) for South African river catchments. Water, 12(6), 1534.

- Guo, Y., Liu, C., Ye, R., & Duan, Q. (2020). Advances on Water Quality Detection by UV-Vis Spectroscopy. Applied Sciences, 10(19), 6874.

- Giudicianni, C., Herrera, M., Di Nardo, A., Greco, R., Creaco, E., & Scala, A. (2020). Topological Placement of Quality Sensors in Water-Distribution Networks without the Recourse to Hydraulic Modeling. Journal of Water Resources Planning and Management, 146(6), 04020030.

- Vonach, T., Tscheikner-Gratl, F., Rauch, W., & Kleidorfer, M. (2018). A heuristic method for measurement site selection in sewer systems. Water, 10(2), 122.

- Yang, S., Paik, K., McGrath, G. S., Urich, C., Krueger, E., Kumar, P., & Rao, P. S. C. (2017). Functional topology of evolving urban drainage networks. Water Resources Research, 53(11), 8966-8979.

- Banik, B. K., Alfonso, L., Di Cristo, C., Leopardi, A., & Mynett, A. (2017). Evaluation of different formulations to optimally locate sensors in sewer systems. Journal of Water Resources Planning and Management, 143(7), 04017026.

- Banik, B. K., Di Cristo, C., Leopardi, A., & de Marinis, G. (2017). Illicit intrusion characterization in sewer systems. Urban water journal, 14(4), 416-426.

In the proposed method there is not reference to the hydraulic conditions of the system; the Authors didn’t carry out hydraulic simulations, neither use specific software for testing the proposed approach. In this regard, I wonder how is it possible to evaluate the efficiency of the method and how reliable are the results. This is a crucial point to discuss for the applicability of the framework in real cases. It seems that the sensor placement problem is addressed without taking into account the physical behaviour of the system that actually is characterised by a daily variability (discharge at each node, dry or rainy conditions, etc…). I strongly recommend to clarify this point;

In order to make the paper more readable and to follow the whole procedure, I suggest to provide a clear general flow-chart of the framework you proposed;

Have you assumed that only a contamination at time can start? Could you discuss all the assumptions behind your method?

Page 16: “It also allows analyzing the structural properties of the sensor location problem including robustness to perturbations in sensor locations”. Could you clarify this point and provide more explanations?

What Qab stand for? What does it represent?

What is the perspective of the future work? I propose to consider the possibility of completing the discussion on possibilities and limitations of the use of such analysis in other networked systems and infrastructures;

Could you also discuss about the computational complexity of your procedure? The scalability and the time requested for the analysis? This is important for the application to real big-sized systems;

Another crucial point is the fact that there is no explicitly reference to the validation of the proposed approach through performance indices especially because it is not tested on real case study neither compared with methods already developed. I strongly recommend to improve this aspect;

It is not clear the reason why the Authors presented a real case study, reconstruct the sewer network with a procedure and then they applied the sensor placement algorithm to another network. Could you clarify these aspects? Furthermore, why didn't you work directly on the real topology of the case study? All these hypotheses about the system could strongly affect the results?

Reviewer #2: The paper “Placing sensors in sewer networks: A system to pinpoint new cases of Coronavirus” presents a method to simulate tree networks, that replicate real sewer pipeline networks, for determining the number of sewage testing required to trace it back to the source of the infection and it addresses the sensors location problem in order to minimize the number of manhole samples required to find the ‘source manhole’. Heuristics are developed to solve this non-linear optimization problem. The content of the paper is a topical issue that may offer points for further research and new operational paths. However, I have several issues (see Major comments below) with the paper and therefore I cannot recommend acceptance in present form. The English usage is not of publication quality and requires major improvement to facilitate reading fluency and comprehension. A few suggestions in this respect are listed in the minor comments below.

MAJOR COMMENTS:

- Page 2 line 9 – “We build on a new field, “Wastewater-Based Epidemiology” (WBE)”. Several strategies in analyzing wastewater to measure biological markers of microbial disease have already been developed and efforts have been made in the field of monitoring sewer systems. At page 28 you state that “the most advanced effort appears to be in Israel, where Ben-Gurion University of the Negev has signed with the Health Ministry to monitor and detect COVID-19 in sewage samples from 14 Israeli communities” and cite the article “Israeli tech to hunt for early evidence of coronavirus in sewer systems” The Jerusalem Post, October 27, 2020. Consider adding a section with a proper state of the art overview, where you consider the major contributions to the field.

- There is no reference to the simulation software you used to test the proposed approach.

- Page 6 line 5 – “According to public works experts […] each test would require about one hour”. It is not clear which type of sewage testing you are referring to and which kind of testing methods are currently available to analyze manhole samples.

- In section 3.2 you call “Q” the ordered list of new pipeline segments to be created, while in section 5.3 you define = log() + log(). Clarify in which way Q and are related or consider using different letters to indicate the two variables.

- Page 9 line 5 – you make a reference to a figure (Figure 1), depicting a simple catchment zone in the interior of a city block, that is not present in the manuscript. Consider including it in the present work or delete the reference.

- Page 19 – “the number of sensors given in (2) is based on two strong assumptions. First, it assumes all manholes have the same Bayesian probability, a useful assumption for obtaining an approximate number of sensors but may require fine-tuning when the Bayesian probabilities are not equal. Second, it assumes that the expected number of manholes can be used, ignoring the probability distribution.” These are strong assumptions that are a bit in contrast with what you stated in the previous sections. How do these simplifying assumptions affect the results?

MINOR COMMENTS:

- Page 8 – “with the terminating street for vehicles often having a “stop sign”.” This sentence risks to confuse the reader, since you are still referring to sewage system tree networks. It is not clear what you mean by this sentence.

- Page 8 – “the 4-intersection has three sewer pipes flowing into a fourth pipe” –> “the 4-segment intersection […]”.

- Page 9 – “The infected “Newbie” individual need not be on the West-East street to the left of this intersection. That individual could be located just north or just south of the intersection.” Reformulate this sentence for a better understanding of its meaning.

- Page 7 – “three types of pipes, each is ascending diameter for increased flows” –> “each has an ascending diameter for increasing flow conditions”.

- Page 7 line 2 – “between 200 to over 5,000 manholes” –> “between 200 and 5,000 manholes” or “from 200 to over 5,000 manholes”.

- In general, the English format is a bit informal, try to avoid expressions like “with say” and rhetorical questions (see page 6 line 3).

6. PLOS authors have the option to publish the peer review history of their article (what does this mean?). If published, this will include your full peer review and any attached files.

Reviewer #1: No

Reviewer #2: No

---

## [Decision Letter · Decision Letter 1]

8 Mar 2021

Placing Sensors in Sewer Networks A System to Pinpoint New Cases of Coronavirus

PONE-D-20-38252R1

Dear Dr. Nourinejad,

We’re pleased to inform you that your manuscript has been judged scientifically suitable for publication and will be formally accepted for publication once it meets all outstanding technical requirements.

Kind regards,

Gabriele Oliva, Ph.D

Academic Editor

PLOS ONE

Additional Editor Comments (optional):

All reviewers agree that the issues have been addressed. I concur with their evaluation and I am recommending acceptance.

Reviewers' comments:

Reviewer's Responses to Questions

**Comments to the Author**

1. If the authors have adequately addressed your comments raised in a previous round of review and you feel that this manuscript is now acceptable for publication, you may indicate that here to bypass the “Comments to the Author” section, enter your conflict of interest statement in the “Confidential to Editor” section, and submit your "Accept" recommendation.

Reviewer #1: All comments have been addressed

Reviewer #2: All comments have been addressed

2. Is the manuscript technically sound, and do the data support the conclusions?

Reviewer #1: Yes

Reviewer #2: Yes

3. Has the statistical analysis been performed appropriately and rigorously? 

Reviewer #1: Yes

Reviewer #2: N/A

4. Have the authors made all data underlying the findings in their manuscript fully available?

Reviewer #1: Yes

Reviewer #2: Yes

5. Is the manuscript presented in an intelligible fashion and written in standard English?

Reviewer #1: Yes

Reviewer #2: (No Response)

6. Review Comments to the Author

Reviewer #1: The authors addressed or give a suitable answer to all the questions I raised in the previous review.

I think the manuscript has been improved significantly.

I recommend it to be accepted for publication in its current version.

Reviewer #2: (No Response)

7. PLOS authors have the option to publish the peer review history of their article (what does this mean?). If published, this will include your full peer review and any attached files.

Reviewer #1: No

Reviewer #2: No

---

## [Editor Report · Acceptance letter]

26 Mar 2021

PONE-D-20-38252R1 

Placing Sensors in Sewer Networks: 
A System to Pinpoint New Cases of Coronavirus 

Dear Dr. Nourinejad:

I'm pleased to inform you that your manuscript has been deemed suitable for publication in PLOS ONE. Congratulations! Your manuscript is now with our production department. 

Kind regards, 

on behalf of

Dr. Gabriele Oliva 

Academic Editor

PLOS ONE